# Quantum Darwinism in a Composite System: Objectivity versus Classicality

**DOI:** 10.3390/e23080995

**Published:** 2021-07-31

**Authors:** Barış Çakmak, Özgür E. Müstecaplıoğlu, Mauro Paternostro, Bassano Vacchini, Steve Campbell

**Affiliations:** 1College of Engineering and Natural Sciences, Bahçeşehir University, Beşiktaş, İstanbul 34353, Turkey; 2Department of Physics, Koç University, Sarıyer, İstanbul 34450, Turkey; omustecap@ku.edu.tr; 3Centre for Theoretical Atomic, Molecular and Optical Physics, School of Mathematics and Physics, Queen’s University Belfast, Belfast BT7 1NN, UK; m.paternostro@qub.ac.uk; 4Dipartimento di Fisica “Aldo Pontremoli”, Università degli Studi di Milano, via Celoria 16, 20133 Milan, Italy; bassano.vacchini@mi.infn.it; 5Istituto Nazionale di Fisica Nucleare, Sezione di Milano, via Celoria 16, 20133 Milan, Italy; 6School of Physics, University College Dublin, Belfield, Dublin 4, Ireland; 7Centre for Quantum Engineering, Science, and Technology, University College Dublin, Belfield, Dublin 4, Ireland

**Keywords:** quantum Darwinism, decoherence, open quantum systems

## Abstract

We investigate the implications of quantum Darwinism in a composite quantum system with interacting constituents exhibiting a decoherence-free subspace. We consider a two-qubit system coupled to an *N*-qubit environment via a dephasing interaction. For excitation preserving interactions between the system qubits, an analytical expression for the dynamics is obtained. It demonstrates that part of the system Hilbert space redundantly proliferates its information to the environment, while the remaining subspace is decoupled and preserves clear non-classical signatures. For measurements performed on the system, we establish that a non-zero quantum discord is shared between the composite system and the environment, thus violating the conditions of strong Darwinism. However, due to the asymmetry of quantum discord, the information shared with the environment is completely classical for measurements performed on the environment. Our results imply a dichotomy between objectivity and classicality that emerges when considering composite systems.

## 1. Introduction

The theory of open quantum systems provides frameworks to describe how an environment destroys quantum superpositions. The environment is an active player in the loss of quantumness and emergence of classicality. In particular, quantum Darwinism [1,2,3] and spectrum broadcast structures [4,5] establish mathematically rigorous approaches to quantitatively assess the mechanisms by which classically objective, accessible states emerge. The former employs an entropic approach, where classical objectivity follows from the redundant encoding of copies of the system’s pointer states (preferred basis states immune to spoiling by system–environment coupling) across fragments of the environment. Objectivity becomes possible only if the information content of the system is redundantly encoded throughout the environment, such that the mutual information shared between the system and arbitrary fractions of the environment is the same and equal to the system entropy. Quantitatively, this implies that the mutual information, given as
(1)$$I\left(\rho_{S:E_{f}}\right)=S\left(\rho_{S}\right)+S\left(\rho_{E_{f}}\right)-S\left(\rho_{S:E_{f}}\right),$$
where S(ρ)=−tr[ρlogρ] is the von Neumann entropy, takes the value $I\left(\rho_{S:E_{f}}\right)=S\left(\rho_{S}\right)$ independently of the size of the environmental fraction $E_{f}$ for f<1, and attains the maximum value of $2S(\rho_{S})$ only when one has access to the whole environment, i.e., f=1. Spectrum broadcast structures approach the problem by rather examining the geometric structure of the state, establishing strict requirements that it must fulfill. Nevertheless, both employ the same notion of objectivity as defined in Ref. [6]:


*“A system state is objective if it is (1) simultaneously accessible to many observers (2) who can all determine the state independently without perturbing it and (3) all arrive at the same result [3,4,7].”*


Recently, the close relationship between quantum Darwinism and spectrum broadcasting structures was established, demonstrating that the latter follows when more stringent conditions, giving rise to the so-called strong quantum Darwinism, are imposed on the former [6,8,9,10].

Establishing whether a given state can be viewed as classically objective has proven to be a difficult task. While any measurement is objective for a classical system, for two or more observers to agree on their measurement results in the quantum case, the basis that they measure must form orthogonal vector sets [11]. Recent studies have shown that the emergence of a redundancy plateau in the mutual information, the characteristic signal of quantum Darwinism, may not imply classical objectivity [4,8,12]. Several critical analyses of quantum Darwinism have demonstrated that, while a generic feature of quantum dynamics [13], it is nevertheless sensitive to seemingly small changes in the microscopic description [14,15], and non-Markovian effects can suppress the emergence of the phenomenon, although the relation between non-Markovianity and quantum Darwinism is yet to be fully understood [16,17,18,19,20,21]. Going beyond the single system particle, the proliferation of system information in spin registers interacting with spin [22] and boson [23] environments have shown to present different characteristics. In addition, while not necessary for decoherence, system–environment entanglement is required for objectivity as defined by quantum Darwinism [24]. Experimental tests of this framework in platforms consisting of photonic systems [25,26,27] and nitrogen-vacancy centers [28] have recently been reported.

This work presents a detailed analysis of redundant information encoding, classical objectivity, and quantum Darwinism for a composite system. We consider two interacting qubits that are coupled to a dephasing environment, as shown in Figure 1. We show that clear Darwinistic signatures are present when the mutual interaction between the two systems is excitation preserving. However, while the composite system establishes precisely the strong correlations necessary for the redundant proliferation of the relevant information, (in this case the system’s total spin), the system establishes a decoherence-free subspace for the dynamics, which is blind to the environmental effects and allows the system to maintain highly non-classical features. We carefully assess whether the redundant information is classical or not by studying the asymmetric quantum discord [29], and demonstrating that the classicality of the mutual information is relative to the observer’s perspective on measuring the system or the environment. For measurements on the system, the state has a significant non-zero quantum discord. Therefore, it violates the conditions set out for objectivity, according to the strong quantum Darwinism criteria [6]. On the other hand, for measurements on the environment side, which are arguably more in the original spirit of quantum Darwinism, the accessible information is completely classical. Our results suggest that the conditions for the objectivity of a system state, as shared by many observers, have to be distinguished from its classicality, understood as the absence of quantum correlations, for the case in which the system has a composite structure. This point turns out to be particularly subtle, as the lack of classicality may not be perceived by an external observer, due to the asymmetry of quantum discord.

The remainder of the manuscript is organized as follows. In Section 2, we introduce details of the composite system model in which we investigate quantum Darwinism. We continue with the behavior of coherences and correlations among the constituents of our model throughout the dynamics, and discuss them in relation to the emergence of Darwinism and objectivity in Section 3. In Section 4, we put forward some considerations on the relation between the phenomenology observed in the case here at hand and the formalism of strong quantum Darwinism, pointing at the asymmetry of quantum discord and its implications for the ascertaining of classicality. In Section 5, we draw our conclusions.

## 2. Composite System Model

We begin by introducing the dynamical model that we consider throughout the paper. Our focus is on exploring how signatures of redundant encoding and objectivity manifest when the system itself is a complex entity with internal interactions among its components. To this end, our system consists of two qubits, $S_{1}$ and $S_{2}$, interacting via the following:(2)$$H_{S_{1}S_{2}}=\sum_{j=x,y,z}J_{j}\sigma_{S_{1}}^{j}⊗\sigma_{S_{2}}^{j}.$$

The composite system interacts with a bath of spins $\left\{E_{k}\right\}$ via a pure dephasing interaction $H_{S_{i}E_{k}}=J_{SE}\sigma_{S_{i}}^{z}⊗\sigma_{E_{k}}^{z}$ with i=1,2 and k=1,2,…,N. A single interaction between any two qubits in the model is realized by the application of the unitary operator $U=e^{-iHt}$ to the state of the system, where $H=H_{S_{1}S_{2}}$ ($H=H_{S_{i}E_{k}}$) for the interactions between $S_{1}$-$S_{2}$ ($S_{i}$-$E_{k}$). We set the initial state of the system and environmental qubits to be a factorized state of the following form:(3)$$|\Psi_{0}\rangle ={|\phi \rangle }_{S_{1}}⊗{|\phi \rangle }_{S_{2}}\overset{N}{\underset{k=1}{⨂}}{|\Phi \rangle }_{k},$$
where ${|\phi \rangle }_{S_{j}}=\cos\theta_{j}{|0\rangle }_{S_{j}}+\sin\theta_{j}{|1\rangle }_{S_{j}}$ (j=1,2), and ${|\Phi \rangle }_{k}={|+\rangle }_{k}=\left({|0\rangle }_{k}+{|1\rangle }_{k}\right)/\sqrt{2}$, with $\left\{|0\rangle ,|1\rangle \right\}$ being the eigenvectors of $\sigma^{z}$ for any of the subsystems involved.

In order to make the model analytically tractable, we enforce the number of interactions with the environment to be uniformly distributed, i.e., both system qubits interact with each environmental qubit in an identical manner, cf. Figure 1. This condition is important since, as demonstrated in Ref. [15], allowing for a bias in the interactions between the system and particular environmental constituents results in a deviation from a Darwinistic behavior that would otherwise be present in the model. Furthermore, by taking the interaction between the system qubits as excitation preserving (i.e., for $J_{x}=J_{y}=J$) the system–system and combined system–environment interaction Hamiltonians commute, i.e., $\left[H_{S_{1}S_{2}},\left(H_{S_{1}E_{k}}+H_{S_{2}E_{k}}\right)\right]=0$. This implies that the ordering of interactions does not matter. Note that when this condition does not hold, the dynamics can still be well simulated by the collisional approach [15,30,31,32]; however, as discussed in Ref. [15], other system–environment interaction terms often lead to a loss in redundant encoding. This simplification allows us to work with the continuous-time *t*, always measured in inverse units of the coupling strength *J*, rather than employing the sequential collisional approach of Ref. [15]. This leads to an analytical expression for the dynamics of the whole *S*-*E* state after time *t*, which we write as follows:(4)$$\begin{array}{ccc} |\Psi \rangle & = & e^{-iNJ_{z}t}\alpha {|00\rangle }_{S_{1}S_{2}}\overset{N}{\underset{k=1}{⨂}}\frac{1}{\sqrt{2}}\left(e^{-i2J_{SE}t}{|0\rangle }_{k}+e^{i2J_{SE}t}{|1\rangle }_{k}\right)\\ & & +e^{iNJ_{z}t}\left[\beta \cos(Jt)-i\gamma \sin(Jt)\right]{|01\rangle }_{S_{1}S_{2}}\overset{N}{\underset{k=1}{⨂}}\frac{1}{\sqrt{2}}\left({|0\rangle }_{k}+{|1\rangle }_{k}\right)\\ & & +e^{iNJ_{z}t}\left[\gamma \cos(Jt)-i\beta \sin(Jt)\right]{|10\rangle }_{S_{1}S_{2}}\overset{N}{\underset{k=1}{⨂}}\frac{1}{\sqrt{2}}\left({|0\rangle }_{k}+{|1\rangle }_{k}\right)\\ & & +e^{-iNJ_{z}t}\delta {|11\rangle }_{S_{1}S_{2}}\overset{N}{\underset{k=1}{⨂}}\frac{1}{\sqrt{2}}\left(e^{i2J_{SE}t}{|0\rangle }_{k}+e^{-i2J_{SE}t}{|1\rangle }_{k}\right) \end{array}$$
with $\alpha =\cos\theta_{1}\cos\theta_{2}$, $\beta =\cos\theta_{1}\sin\theta_{2}$, $\gamma =\sin\theta_{1}\cos\theta_{2}$, $\delta =\sin\theta_{1}\sin\theta_{2}$. Immediately we see some tell-tale signatures of Darwinism appearing: $|00\rangle$ and $|11\rangle$ states imprint the same type of phase on the environmental qubits as shown in Ref. [15]. We see that in the single excitation subspace of *S*, while the mutual interaction only exchanges populations, the environmental qubits are not affected. In what follows, we demonstrate that these features conspire to complicate the decision on whether a classically objective state has been achieved or not: the state in Equation (4) exhibits clear signatures of redundant encoding in the environment while allowing the system to maintain highly non-classical features within a subspace that the environment is, in effect, “blind” to.

## 3. Quantum Darwinism and Objectivity

Quantitatively, quantum Darwinism is signaled by a plateau in the mutual information shared between the system and a fraction of the environment at the entropy of the system’s state plotted against the fraction size. This behavior is indicative of a redundant encoding of the system information throughout the environment such that, regardless of what fragment of the environment is queried, an observer only ever has access to the same information. While there can be a “minimum fragment" size necessary to reach the redundancy plateau [3], we focus on the extreme case where it is sufficient to query a single environmental qubit in order to obtain all the accessible information. This amounts to tracking the mutual information $I(\rho_{S_{1}S_{2}:E_{k}})$ between the composite two-qubit system and a single environmental qubit together with the entropy of the former $S(\rho_{S_{1}S_{2}})$, such that $I\left(\rho_{S_{1}S_{2}:E_{k}}\right)=S\left(\rho_{S_{1}S_{2}}\right)$ indicates we are witnessing a classically objective state, according to quantum Darwinism. Unless stated otherwise, we fix the mutual interaction between the qubits to be $H_{S_{1}S_{2}}=J\left(\sigma_{x}⊗\sigma_{x}+\sigma_{y}⊗\sigma_{y}\right)$ and the system–environment coupling is $J_{SE}=0.1J$ to consider conditions of weak system–environment coupling. The system qubits are assumed to be initially prepared in identical states with $\theta_{1}=\theta_{2}=\pi /6$. This choice is only dictated by the convenience of the illustration of our results and, aside from some minor quantitative differences, qualitatively similar results hold for any choice or combination of *J* and $J_{z}$, including non-interacting system qubits, i.e., $J=J_{z}=0$, and also for different initial system states. Finally, we note that, as we assume uniform coupling to all environmental units, it is immaterial which is chosen in the evaluation of $I(\rho_{S_{1}S_{2}:E_{k}})$.

Figure 2a shows the mutual information, $I(\rho_{S_{1}S_{2}:E_{k}})$ and the composite system entropy $S(\rho_{S_{1}S_{2}:E_{k}})$ for environments consisting of N=6 (solid) and N=250 qubits (dashed). We immediately see that only at t=π/4 do we find $I\left(\rho_{S_{1}S_{2}:E_{k}}\right)=S\left(\rho_{S_{1}S_{2}}\right)$. For N=250, the system entropy quickly saturates to a maximum value, which is dependent on the chosen initial states, and remains so for most of the dynamics with the notable exception of t=π/4, where it rapidly drops. A qualitatively identical behavior is exhibited by the coherence present in the two-qubit system state, $C=\sum_{i\neq j}\left|\rho_{S_{1}S_{2}}^{i,j}\right|$, shown in Figure 2b. While each system qubit quickly becomes diagonal, remarkably, the composite system maintains a minimum value of coherence, indicating that it retains some genuine non-classicality, with the magnitude of the coherence being dependent on the particular choice of initial states for the systems.

The mutual information shared between the system qubits and an environment, shown in Figure 2a (darker, black curves), varies more gradually and is inversely related with the behavior of the environmental qubit’s coherence, $|\rho_{E_{k}}^{1,2}|$, shown in Figure 2b (darker, black curve). The point at which $I\left(\rho_{S_{1}S_{2}:E_{k}}\right)=S\left(\rho_{S_{1}S_{2}}\right)$ corresponds to the minimum in the environment coherence establishing that in order for signatures of objectivity to emerge a mutual dephasing is necessary [15]. Nevertheless, Figure 2b is remarkable, as it indicates that we do not require all constituents to become fully classical, i.e., the coherence does not necessarily vanish. In Figure 2d, the solid lines show the mutual information between the composite system and fractions of the environment at t=π/4 and t=(π/4−0.1) for a N=6 qubit environment. When t=(π/4−0.1), there are no clear indications of Darwinistic behavior, while at t=π/4, we clearly observe the characteristic plateau, indicating that the system information is redundantly encoded into the environmental degrees of freedom. From these results, we see that the presence of a redundancy plateau does not necessarily imply a complete loss of all non-classicality within a complex composite system.

We can examine these features more quantitatively by directly computing the reduced and the total states of the system for t=π/4. The density matrix for $S_{1}S_{2}$ is *X*-shaped, which in turn enforces the reduced states to be diagonal as follows:$$\rho_{S_{1}S_{2}}=\left(\begin{array}{cccc} \alpha^{2} & 0 & 0 & \alpha \delta \\ 0 & \beta^{2} & \beta \gamma & 0\\ 0 & \beta \gamma & \gamma^{2} & 0\\ \alpha \delta & 0 & 0 & \delta^{2} \end{array}\right),\;$$
$$\rho_{S_{1}}=\left(\begin{array}{cc} \alpha^{2}+\beta^{2} & 0\\ 0 & \gamma^{2}+\delta^{2} \end{array}\right),\;\rho_{S_{2}}=\left(\begin{array}{cc} \alpha^{2}+\gamma^{2} & 0\\ 0 & \beta^{2}+\delta^{2} \end{array}\right).$$

Therefore, the non-zero coherence we see in Figure 2 when the Darwinistic plateau emerges can be analytically determined to be $\sum_{i>j}|\rho_{S_{1}S_{2}}^{i,j}\left|=|\alpha \delta |+|\beta \gamma \right|$. The coherence contained in a single environmental qubit $|\rho_{E_{k}}^{1,2}|$ is dependent on the initial states of the system qubits but independent with regards to the overall size of the environment, *N*. In particular, when t=π/4, we find $|\rho_{E_{k}}^{1,2}|=\left(\beta^{2}+\gamma^{2}-\alpha^{2}-\delta^{2}\right)/2$, indicating that the environmental qubits themselves will fully decohere only when either $\theta_{1}$ or $\theta_{2}=\pi /4$, while for all other values of initial states some non-classicality remains within the environmental constituents.

Since the composite system maintains non-zero coherence, even when a redundancy plateau is observed, it is relevant to examine any non-classical correlations present in the overall state, cfr. Figure 2c, where we show the quantum discord [33,34] between two system qubits $D^{\to }\left(\rho_{S_{1}S_{2}}\right)$ and quantum discord between one of the system particles and a single environment $D^{\to }\left(\rho_{S_{1}E_{k}}\right)$. Mathematically, quantum discord between two parties is defined as [33,34]
(5)$$D^{\to }\left(\rho_{AB}\right)=I\left(\rho_{AB}\right)-J^{\to }\left(\rho_{AB}\right).$$

Here, $J^{\to }\left(\rho^{AB}\right)=S\left(\rho^{B}\right)-\min_{\left\{\Pi_{k}^{A}\right\}}\sum_{k}p_{k}S\left(\rho_{k}^{B}\right)$ is called the Holevo information, where {$\Pi_{k}^{A}$} represents the set of all possible measurement operators that can be performed on subsystem *A*, and $\rho_{k}^{B}=\left(\Pi_{k}^{A}⊗I\right)\rho^{AB}\left(\Pi_{k}^{A}⊗I\right)/p_{k}$ are the post-measurement states of *B* after obtaining the outcome *k* with probability $p_{k}=\mathrm{tr}\left[\left(\Pi_{k}^{A}⊗I\right)\rho^{AB}\right]$. In other words, $J^{\to }\left(\rho_{AB}\right)$ measures the amount of information that one can obtain about subsystem *B* by performing measurements on subsystem *A*. The non-zero coherence present in the $S_{1}S_{2}$ state mean that there are genuine quantum correlations in the form of the discord shared between the system qubits and, despite exhibiting a sharp decrease near the point where the characteristic plateau emerges, they remain non-zero throughout the dynamics. Thus, it is natural to question whether we can consider the state as truly objective when the relevant system information has clearly proliferated into the environment. Examining the correlations established between a given system qubit and one environmental constituent, $D^{\to }\left(\rho_{S_{1}E_{k}}\right)$, we find the quantum discord vanishes when the redundancy plateau is observed, implying that, at least at the level of a single system constituent, only classical information is accessible. We thus have a situation in which, due to the presence of a decoherence-free subspace to which the environmental degrees of freedom is blind, the overall composite system maintains non-classical features, and therefore, is arguably not objective, despite the redundant encoding and proliferation of the system information. Such a situation is reminiscent of settings where solely focusing on the mutual information can provide a false flag for classical objectivity [8]. Therefore, in the following section, we turn our attention to tighter conditions for objectivity given by strong quantum Darwinism [6,8], or equivalently spectrum broadcast structures [4,5].

Before moving on, we believe it is also meaningful to explore whether Darwinistic signatures are exhibited when considering how much information the environment can access about the individual system constituents, i.e., whether in addition to checking $I\left(\rho_{S_{1}S_{2}:E_{f}}\right)=S\left(\rho_{S_{1}S_{2}}\right)$ for various fragment sizes, we also test whether $I\left(\rho_{S_{i}:E_{f}}\right)=S\left(\rho_{S_{i}}\right)$. We note, however, already that the latter quantity is upper bounded by the former, i.e., $I\left(\rho_{S_{1}S_{2}:E_{f}}\right)\geq I\left(\rho_{S_{1}:E_{f}}\right)$ for all $E_{f}$ and at all times since discarding a system never increases the mutual information, due to the strong subadditivity of von Neumann entropy [35].

Figure 2d shows $I(\rho_{S_{1}S_{2}:E_{f}})$ (solid) and $I(\rho_{S_{1}:E_{f}})$ (dashed) against the fraction size of the environment *f* for t=π/4 (red) and t=(π/4−0.1) (blue). Focusing on t=π/4, it is clear that the composite system exhibits the characteristic redundancy plateau with the mutual information exactly equaling the composite system entropy. While we observe a similar plateau for the mutual information between a reduced system state and environment fractions, in this case, it is below the entropy of the considered system particle. Therefore, the redundantly encoded information regarding the single system qubit does not contain the full information about the qubit itself, and this is due to the fact that some of the information—specifically, that which is tied up in the non-classical correlations shared between the two system qubits—is not classically accessible to the environment.

The discrepancy between $I(\rho_{S_{1}S_{2}:E_{f}})$ and $I(\rho_{S_{1}:E_{f}})$, which is related to the gap between the two curves in Figure 2d, can be quantitatively determined by directly computing their difference $\Delta I=I\left(\rho_{S_{1}S_{2}:E_{f}}\right)-I\left(\rho_{S_{1}:E_{f}}\right)$ and finding the following (see Appendix A):(6)$$\Delta I=I\left(\rho_{S_{2}:E}\right)-I\left(\rho_{S_{2}:{\bar{E}}_{f}}\right),$$
where ${\bar{E}}_{f}$ is the complement of $E_{f}$. Note that this result is completely independent of the nature of the dynamics and valid for arbitrary fractions at any given instant, only relying on the assumption of a pure initial state. Furthermore, another useful insight regarding this mutual information gap can be provided by exploiting the Koashi–Winter relation [36], which helps us to bound the discrepancy as follows:(7)$$0\leq \Delta I\leq S\left(\rho_{S_{1}S_{2}}\right)+D^{\leftarrow }\left(\rho_{S_{1}S_{2}:E_{f}}\right).$$

Considering the fraction to be the whole environment, i.e., ${\bar{E}}_{f}$ is an empty set, the expressions above reduce to the following simple form $\Delta I=I(\rho_{S_{2}:E})$, which corresponds to the gap at the end of the curves. Equations (6) and (7) demonstrate that the non-classical correlations present in the composite system prevent the environment from gaining complete and unambiguous information regarding the state of an individual subsystem (see Appendix A for more details).

## 4. Strong Quantum Darwinism

The previous section demonstrates that, despite observing the signature plateau for redundant encoding, the constituents of the model still carry certain signatures of quantumness, namely non-zero coherences and discord of the composite system state. This naturally leads us to question of whether one can argue that the system state is truly classically objective or not. While it is clear that there is a proliferation and redundant encoding of system information within the environment, the fact that the system itself persists in displaying non-classical features implies that there might be a subtle distinction between the proliferation of relevant system information and genuine classical objectivity of a quantum state, with the former being a necessary but not sufficient requirement for the latter. Such a critical analysis of the conditions for classical objectivity is formalized within the framework of spectrum broadcast structures and strong quantum Darwinism [4,8,12]. In particular, the *strong Darwinism* condition [6,9] amounts to determining whether or not the mutual information shared between the system and an environment fraction is purely composed of classical information as quantified by the Holevo information. Equivalently, this condition can be stated as whether or not the system has a vanishing discord with that environment fraction. While stated originally based on measurements performed *solely on the system*, this condition was shown to be a necessary and sufficient condition for classical objectivity [6]. However, it is known that quantum discord is an asymmetric quantity, dependent on precisely which subsystem is measured and, therefore, allows for curious situations where non-classical correlations can be shared in one direction but not the other, so-called quantum-classical states. Therefore, even though the framework of strong Darwinism established in Ref. [6] is well motivated, we argue that, while objectivity is a property based on information that can be accessed by measurements on the *environment only*, classicality is a more subtle issue, and due to the asymmetry of quantum discord, a system state, though assessed as classically objective from the environment or a fraction of it, can retain quantum correlations [37].

The calculation of quantum discord is involved, even for two-qubit states [29]; in fact, it can be shown to be a NP-complete problem [38]. However, for our purposes, it suffices simply to check whether or not there exists discord without computing its numerical value. Thus, we focus on the correlation properties shared between the composite system and a single environment, taking into account the asymmetric nature of the discord. We employ a nullity condition, which provides a necessary and sufficient condition to witness whether the state, $\rho_{S_{1}S_{2}:E_{k}}$, has zero discord [39,40,41]. An arbitrary state $\rho_{AB}$ has a vanishing quantum discord with measurements on *A* or *B* if and only if one can find an orthonormal basis $\left\{|n\rangle \right\}$ or $\left\{|m\rangle \right\}$ in the Hilbert space of *A* or *B*, such that the total state can be written in a block-diagonal form in this basis. Mathematically, it is possible to express this condition as follows:(8)$$\begin{array}{cc} D^{\to }\left(\rho_{A:B}\right)=0 & \Leftrightarrow \rho_{AB}=\sum_{n}p_{n}|n\rangle \langle n|⊗\rho_{n}^{B}, \end{array}$$(9)$$\begin{array}{cc} D^{\leftarrow }\left(\rho_{A:B}\right)=0 & \Leftrightarrow \rho_{AB}=\sum_{m}q_{m}\rho_{m}^{A}⊗|m\rangle \langle m|. \end{array}$$

Note that in our case, we make the identification $A\to S_{1}S_{2}$ and $B\to E_{1}$. In Appendix B, we explicitly calculate a necessary and sufficient condition for the nullity of quantum discord introduced in [40,41], separately considering both cases of measurements performed on the system and the environment side. When the mutual information plateau is observed while $D^{\to }\left(\rho_{S_{1}S_{2}:E_{1}}\right)$ is non-zero, for measurements performed in the environmental qubit, the discord $D^{\leftarrow }\left(\rho_{S_{1}S_{2}:E_{1}}\right)$ vanishes. Thus, we have a quantum–classical state implying that, as far as measurements are only performed on the environment, all the accessible information is completely classical in nature. As already discussed, this result implies an important subtlety regarding the connection between quantum Darwinism and the emergence of objectivity or classicality in composite quantum systems. While, locally, both system qubits are completely decohered, the composite state of the system is still coherent and shares some non-classical correlations with the environment. It is, thus, non-classical from the perspective of the system. However, from the perspective of the environment, the system is both objective, as the accessible information about the composite system is redundantly encoded throughout its degrees of freedom, and classical in that quantum discord for the measurement performed on the environment is equal to zero.

## 5. Conclusions

We have examined the emergence of quantum Darwinism for a composite system consisting of two qubits interacting with a *N*-partite bath. For an excitation-preserving interaction between the system qubits, we established that the system information is faithfully, redundantly encoded throughout the environment; therefore, we see the emergence of clear Darwinistic signatures. Nevertheless, a decoherence-free subspace permits the system to create and maintain significant non-classical features in the form of quantum discord. Employing the framework of strong quantum Darwinism, which insists that in addition to a mutual information plateau, the discord between the system and an environment fragment must vanish, we have shown that whether or not this state is interpreted as objective and classical depends on how the discord is evaluated. Following the framework of Ref. [6], for measurements on the system, the sizable non-zero coherence present in the decoherence-free subspace implies that this state is definitively not objective. However, as quantum Darwinism posits that classicality and objectivity are dictated by what information can be learned by measuring the environment, and due to the asymmetric nature of the quantum discord when measurements are made on the environment, we find that the discord is vanishing and therefore conclude a classically objective state. To better understand this point, we demonstrated that redundant encoding at the level of the composite system does not imply the same for the individual constituents. Specifically, when non-classical correlations are established between the system qubits, there is still a redundant proliferation of *some* of the system information into the environment; however, the correlations between the two system qubits prevent all of the system’s information from being redundantly shared with the environment.

## Figures and Tables

**Figure 1 entropy-23-00995-f001:**
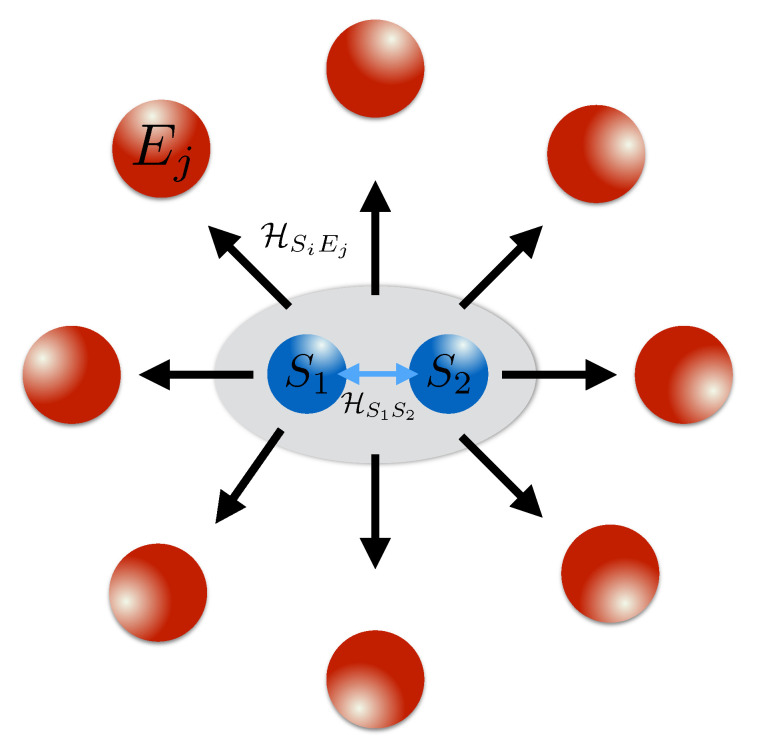
Schematics of the considered model. A composite system that is made up of two interacting qubits, which are also coupled to a fragmented environment. For a excitation preserving interaction between the system qubits, i.e., $J_{x}=J_{y}$ in Equation (2) and pure dephasing interaction between the system qubits and the environment, the interaction Hamiltonians commute, leading to Equation (4).

**Figure 2 entropy-23-00995-f002:**
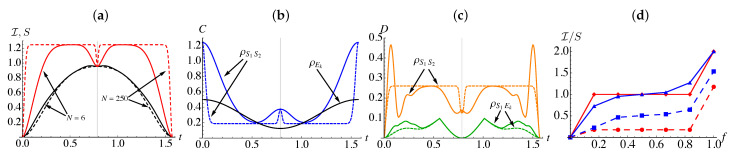
Both system qubits are prepared in a symmetric initial state with $\theta_{S_{1}}=\theta_{S_{2}}=\pi /6$ with the interaction parameters $J_{SE}=1$, J=10, and we consider environments of size N=6 (solid curves) and N=250 (dashed curves). (**a**) Dynamics of mutual information between the composite system and a single environmental qubit, $I(\rho_{S_{1}S_{2}:E_{k}})$, (darker, black) and the entropy of the composite system, $S(\rho_{S_{1}S_{2}})$ (lighter, red). (**b**) Coherence present in the composite system state, $|\rho_{S_{1}S_{2}}^{1,2}|$ (lighter, blue) and coherences in the state of a single environmental qubit $|\rho_{E_{k}}^{1,2}|$ (darker, black). (**c**) Quantum discord shared between the two system qubits, $D^{\to }\left(\rho_{S_{1}S_{2}}\right)$ (lighter, orange) and quantum discord between one of the system qubits and a single environmental constituent $D^{\to }\left(\rho_{S_{1}E_{k}}\right)$ (darker, green). In panels (**a**–**c**) the faint vertical line at t=π/4 denotes the time at which we have $I\left(\rho_{S_{1}S_{2}:E_{k}}\right)=S\left(\rho_{S_{1}S_{2}}\right)$, i.e., the emergence of Darwinism. (**d**) $I\left(\rho_{S_{1}S_{2}:E_{f}}\right)/S\left(\rho_{S_{1}S_{2}}\right)$ vs. the size of the environment fraction *f* (upper, solid) and $I\left(\rho_{S_{1}:E_{f}}\right)//S\left(\rho_{S_{1}}\right)$ (lower, dashed) at two instants of time, t=π/4 where perfect redundant encoding is observed (lighter, red) and t=π/4−0.1 (darker, blue).

## Data Availability

Not Applicable.

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
