# Peer review of "Quantum Darwinism in a Composite System: Objectivity versus Classicality"

_entropy, 2021, doi:10.3390/e23080995_

Round 1

Reviewer 1 Report

In this work, the authors described how a composite system establishes the objectivity in the environment through quantum Darwinism. An interesting feature is that the system has a decoherence-free subspace that decoupling from the environment, thus that they can observe an asymmetric quantum discord between the system and the environment. This result implied that the objectivity and the classicality is not the same concept. This work demonstrates a new feature in quantum Darwinism for composite systems, which is not present in single-particle system. So, I recommend its publication in Entropy.

Author Response

We thank the referee  for their positive appraisal of our work.

Reviewer 2 Report

I find the toy model studied by the authors to be simple enough to let them make their point clearly. Quantum Darwinism appears to be an important notion in recent developments of the theory of quantum open systems, and addressing it through the mutual information allows for a robust conclusion on its emergence in different contexts, depending on the system-environment interaction type. I therefore find that this well written manuscript deserves to be communicated to the reader of Entropy.

Author Response

(The authors gave the same response as above.)

Reviewer 3 Report

This paper provides a nice, clear, simple example which suggests a certain refinement in the formulation of Quantum Darwinism. One cannot speak vaguely about "the" information in a quantum system being recorded redundantly in the environment, because it can easily be that some system information is broadcast while some other system information is not. I think that this point is important enough to make, and is made clearly enough by this paper, that this paper should be published.

The presentation of this point should be improved in several ways, however.

1) First of all the important terms should be clearly defined at the beginning of the paper. Quantum Darwinism, spectrum broadcasting structures, objectivity, and classicality should be defined and explained, so that readers who are not already familiar with the exact meanings of these terms can follow the argument. The present wording about "quantitatively assess mechanisms" in line 19 just seems meaningless to me; we need something clearer than this.

2) Secondly the presentation of the model Hamiltonian could be clarified greatly by avoiding any discussion of "collision-like models" and time-dependent Hamiltonians, since these have nothing to do with what is actually done in this paper. It is quite enough just to present the Hamiltonian and observe that s^z_{S1} + s^z_{S2} is conserved, and couples to \sum_k s^z_{Ek}, which is also conserved, so that the total time evolution is very simple, allowing for example the exact solution (4) for any t.

Once the paper comes to quantitative results I find that the clarity improves greatly. So I only have a few other comments about specific problems.

3a) An easily fixed but important point for clarity: every mention in the paper of the interaction between the system spins being "energy-preserving" must be replaced with something like "total spin conserving", because what is conserved is s^z_{S1} + s^z_{S2}. This is not "energy" in any obvious sense; the energy is the total Hamiltonian, and this is automatically conserved whenever the Hamiltonian is not explicitly time-dependent.

3b) In line 29 it should be clarified that I is independent of the size of the environmental fraction as long as the fraction is less than 100%.

3c) In line 43 the expression "measurements made in orthogonal bases" was confusing to me. Usually the eigenstates of any observable are an orthogonal basis. Do the authors perhaps mean to say "measurements made in separate subspaces" or something like that?

3d) It is a bit annoying that the caption to Figure 1 refers to equations which have not yet been seen because the only appear after Figure 1 in the paper.

3e) On line 122 I believe that "conjure" should instead be "conspire".

3f) On line 324 it is not clear to which information is meant by "all this information".  "All of the system information" might be what the authors mean; if so, they should say this.

3g) At some early point the authors should explain that their basic result is really quite straightforward: the environment only couples to the total spin of the system, which is also conserved by the system Hamiltonian, so there is a decoherence-free subspace spanned by |01> and |10>. So information about the system's total s^z is broadcast redundantly to the environment, but information about the components of the system state vector within the decoherence-free subspace is not recorded at all by the environment.

There is no need to be embarrassed by how simple this result is. On the contrary it is nice to have such a simple example. Once we can see clearly from this simple case that not all the quantum information in a system has to be broadcast equally to the environment, it is easy to imagine that more general systems and environments will still show this effect in some form, in that there will be different degrees of objectivity or classicality for different aspects of the system.

Author Response

We thank the referee for the positive appraisal of our work and for providing some constructive suggestions to improve our manuscript. In what follows we provide a detailed response to each of their comments:

  1. We appreciate the referee's comment, however, feel that our introduction does faithfully define all the relevant concepts. After the offending clause in line 19, lines 20 - 40 are entirely dedicated to defining quantum Darwinism, objectivity, and spectrum broadcast structures, where for brevity we make reference to the most relevant literature.
  2. We see the referee's point, for the specific interaction term studied here one does not need to resort to a collisional picture. However we feel it is important to stress that should one choose any other interaction term, as embodied by Eq. (2), then due to the non-commutativity between the terms, it would still be possible to study the dynamics via a collision model. In order to improve the flow we have relegated this observation to a footnote. 
  3. (a) We have changed "energy preserving" to "spin preserving" as the referee suggests
    (b) We have clarified this statement in line with the referee's suggestion.
    (c) We have clarified this statement in line with the referee's suggestion.
    (d) We appreciate the referee's opinion, however, as the figure is essentially stand alone we feel justified in referring to equations confident that the reader will be able to locate them and understand the schematic more completely.
    (e) We take the referee's point. 
    (f) We take the referee's point.
    (g) We thank the referee for their insight and have added this to the introduction.